# Production and Characterization of Molecular Dications: Experimental and Theoretical Efforts

**DOI:** 10.3390/molecules25184157

**Published:** 2020-09-11

**Authors:** Stefano Falcinelli, Marzio Rosi

**Affiliations:** 1Department of Civil and Environmental Engineering, University of Perugia, Via G. Duranti 93, 06125 Perugia, Italy; 2SCITEC, CNR, Via Elce di Sotto 8, 06123 Perugia, Italy

**Keywords:** molecular dications, Coulomb explosion, mass spectrometry, electron-ion-ion coincidence technique, molecular beam, ion escape, planetary atmospheres, astrochemistry, kinetic energy released, potential energy surfaces, ab initio calculations

## Abstract

Molecular dications are doubly charged cations of importance in flames, plasma chemistry and physics and in the chemistry of the upper atmosphere of Planets. Furthermore, they are exotic species able to store a considerable amount of energy at a molecular level. This high energy content of several eV can be easily released as translational energy of the two fragment monocations generated by their Coulomb explosion. For such a reason, they were proposed as a new kind of alternative propellant. The present topic review paper reports on an overview of the main contributions made by the authors’ research groups in the generation and characterization of simple molecular dications during the last 40 years of coupling experimental and theoretical efforts.

## 1. Introduction

Dications are doubly positive charged atomic or molecular ions. They are formed whenever a chemical system has a high amount of energy available, as for example in flames, and we can find them in low concentrations in many natural plasmas such as planetary ionospheres, interstellar clouds and comet tails [1,2,3]. Indeed, astronomers found ionized gas containing H_2_O^+^, CO^+^, CN^+^, and CO^2+^ molecular dication in the tail of Hyakutake comet in 1996. Then, N_2_^2+^ and CO_2_^2+^ were predicted to exist in the ionosphere of Mars, Venus and Titan (the largest moon of Saturn) where they have been found with significant densities [4,5,6]. Therefore, an interesting review paper was published highlighting the role of dications in the upper atmosphere of planets [7], indicating the possibility for such ionic species to be involved in the escape processes by Coulomb explosion [8,9]. More in general, for an overview of the main characteristics and on the importance of molecular dications, the reader can refer to the relevant review papers by Herman [10], Sabzyan et al. [11] and Price et al. [12].

Considering diatomic dications AB^2+^, a simple way to rationalize their stability is by the use of the Δ parameter, introduced for the first time by Bates and Carson in 1955 [13]. According to the simple Equation (1) below, Δ is defined as the difference between the atomic second ionization potential I_2_ of A, which corresponds to the atom of the pair having the lowest electronegativity, and the first ionization potential I_1_ of the other atom B:Δ = I_2_(A) − I_1_(B).(1)

From Equation (1), it is easy to verify that negative values of Δ lead to the A^2+^ + B potential energy curve lying below A^+^ + B^+^, indicating the thermodynamic stability of AB^2+^ (see Figure 1a). Large positive values of Δ are related to the opposite configuration where the ground state is at best metastable and the dissociation asymptote giving A^+^ + B^+^ lies below the local potential energy minimum of AB^2+^ (see Figure 1b). Finally, small positive values of Δ may correspond to either cases, depending on the depth of the potential well corresponding to the A^2+^ + B configuration (see Figure 1c).

From these arguments, thermodynamically stable diatomic dications are expected to be formed between atoms with low first and second ionization potentials (i.e., divalent metal atoms, like the alkaline earths) and atoms with large first ionization energies, such as the noble gases and the halogens. On the basis of this rationalization, and according to the relative potential energy curves scheme of Figure 1, we can summarize saying that, in general, doubly charged molecular dications are usually unstable species, because of their high formation energy. In the gas phase, the two charged parts of the dication tend to repel each other, making this species metastable, when Δ ≥ 0 with a subsequent avoided crossing between the two potential energy curves describing the A^2+^ + B and A^+^ + B^+^ interactions, and more often unstable (when Δ > 0), determining the fragmentation in two positive ions by the so called Coulomb explosion often with a high kinetic energy release (KER) as depicted in Figure 1b,d.

Historically, the study of diatomic dications has not followed the logic presented above. The existence of doubly charged ions was first suggested either by Thompson [14] and Conrad [15] by the discovery of He^2+^ dications in their discharge experiments, as cited by Aston in his pioneering work on mass spectrometry [16]. Their identification was achieved first by Vaugham [17] with the discovery of CO^2+^ in 1931 and its characterization in term of the ionization energy appearance’s threshold. During the same period, the unexpected stability of molecular dications has been investigated extensively also from a theoretical point of view, and in 1933, Linus Pauling was the first carrying out valence bond calculations [18]: The simplest diatomic dication, He_2_^2+^, was characterized as a metastable species having a well depth of about 1.28 eV, supporting four vibrational levels, with an extremely long lifetime (of about 220 min for the v = 0 level), with an energy content exceptionally high (10.15 eV), which is easily releasable as KER of the two He^+^ fragments formed by Coulomb explosion [18]. These important characteristics, which are summarized in Figure 2, allowed Nicolaides [19] to discuss new types of fuels involving the generation of propulsive energy via the following processes (see Equations (2)−(10) below):

Unimolecular fragmentation induced by a CO_2_ laser or other beams and which can proceed as a chain reaction:He_2_^2+^(^1^Σ_g_, v = 0) + 1.28 eV (≈ 9710 Å) → 2He^+^ + 10.15 eV(2)

Chemical reactions involving light species:He_2_^2+^ + H → HeH^+^ + He^+^ + 20.86 eV,(3)He_2_^2+^ + H → He_2_^+^ + H^+^ + 22.03 eV,(4)He_2_^2+^ + H^−^ → He_2_^+^ + H + 34.86 eV,(5)He_2_^2+^ + H^−^ → HeH^+^ + He + 44.75 eV,(6)He_2_^2+^ + H_2_^+^ → HeH^+^ + He^+^ + H^+^ + 18.13 eV,(7)He_2_^2+^ + H_2_^+^ → He_2_^+^ + 2H^+^ + 19.25 eV,(8)He_2_^2+^ + H_2_^+^ + e^−^ → 2HeH^+^ + 43.88 eV,(9)He_2_^2+^ + H_2_ → He_2_^+^ + H_2_^+^ + 20.08 eV,(10)He_2_^2+^ + H_2_ → 2HeH^+^ + 28.45 eV.(11)

It has to be noted that the exothermicity of reactions (2)–(11) is exceptionally high in comparison to the well-known gas-phase reaction at 25 °CH_2_ + 1/2O_2_ → H_2_O + 2.52 eV.(12)

As many metastable molecular dications are able to release a lot of energy as KER of the fragmentation products formed by Coulomb explosion, they have attracted the attention of the scientific community as possible species able to store energy at a molecular level and to be used as alternative propellants [19,20]. During the years, a number of researchers devoted their scientific activity to study such interesting ionic species. 

From a theoretical point of view, after the pioneering work of Pauling on He_2_^2+^ mentioned above [18], historically, we can cite the following important contributions: Hurley and Maslen [21,22] with the development of their approximate procedure able to estimate the potential energy curves of various metastable dications such as F_2_^2+^, O_2_^2+^, N_2_^2+^, CO^2+^, NO^2+^, and NO_2_^2+^; Boldyrev and Simons [20] with their ab initio studies of metastable multiprotonated species. Later on, high level ab initio calculations by: (i) Radom and coworkers on NF^2+^, CNe^2+^ [23], demonstrating, in the case of O_2_^2+^, the failure of Møller–Plesset perturbation theory [24]; (ii) Wright and coworkers on metastable B_2_^2+^ [25] and then on thermodinamically stable dications [26]; (iii) Miller et al. with their calculations on HS^2+^ and its experimental observation via electron impact ionization of H_2_S [27]; (iv) Levasseur et al. on CO^2+^ [28]; (v) Senekowitsch et al. on F_2_^2+^, OH^2+^, and N_2_^2+^, extending their analysis to describe the general features of the chemical bond in molecular dications following the Pauling’s intuition [29,30]; (vi) Bauschlicher and coworkers in their theoretical description of the bonding in the molecular dications containing transition-metals [31,32]; (vii) Cederbaum and coworkers in the characterization of the dicationic states of hydrocarbons using a statistical approach to their Auger spectra [33,34]; (viii) Hochlaf et al. [35,36] in the characterization of CO_2_^2+^; and (ix) de Melo and Ornellas with their recent interesting characterization of the thermodynamic stable dications containing alkaline earth atoms [37,38,39].

Experimentally, after the first determination by Vaughan [17], main studies were performed using the mass spectrometry technique, as in the case of the production of: (i) NO^2+^, CO^2+^, and N_2_^2+^ by Dorman and Morrison in 1961 [40]; (ii) the thermodynamically stable XeNe^2+^ dication studied in a drift-tube mass spectrometer by Johnsen and Biondi in 1979 [41] and then by Helm et al. [42] who produced also ArXe^2+^ by double ionization of van der Waals dimers; (iii) HS^2+^ and NF^2+^ by Leone and coworkers [26,43]; (iv) SiF^2+^ produced by Brion et al. [44,45] in the photoionization of SiF_4_, that a few years later Kolbuszewski and Wright [46] demonstrated to be thermodynamically stable by multireference CI ab initio calculations. Starting from the 80s, also spectroscopic characterization of N_2_^2+^ and NO^2+^ were done by Cossart et al. [47,48,49].

An important experimental effort was done by Chatterjee and Johnsen [50] in 1989 who presented a study on the reactivity of O_2_^2+^ dications with neon atoms and various simple molecules (N_2_, O_2_, NO, CO_2_) by drift-tube mass-spectrometry. Later, in 1995, van der Kamp et al. [51] were able to perform the first charge-transfer reaction between alkali atoms and both CO^2+^ and NO^2+^ dications.

More recently, it deserves mention the interesting work by Franzreb et al. who were able to produce a variety of oxygen-containing diatomic dications XO^2+^ (X = As, Ga, Sb, Ag, Cr, Be) in the gas phase [52], the metastable ClO^2+^ and ClO^3+^ ions [53], and the thermodynamically stable SrO^2+^ and SrH^2+^ [54]

Finally, it has to be pointed out the important contribution provided by J.H.D. Eland’s research group in the last decades, which has studied and characterized a large number of molecular dications of increasing complexity through the use of the double photoionization combined with electron-ion-ion coincidence techniques [55,56,57,58].

Remarkable recent developments have been done in the production and characterization of multiply ionized molecular species using high-intensity laser experiments, as intense femtosecond lasers and X-ray free electron lasers (see for example references [59,60,61] and references therein). In particular, Yatsuhashi and Nakashima [61] in their recent review paper pointed out important advances performed on Coulomb explosion imaging experiments. This kind of technique allowed the determination of the static structures for complex and chiral molecules, geometric isomers and molecular aggregates. Furthermore, the importance of Coulomb explosion of solids (i.e., metals, dielectrics, semiconductors, polymers, and molecular aggregates) has grown in recent years and pioneering time-resolved studies of surface electric fields have been done clarifying the contribution of Coulomb explosion to the mechanism for ablation of solid surfaces [61].

In the present topic review article, the authors present an overview of the main contributions made by their research groups in the generation and characterization of simple molecular dications during the last 40 years coupling experimental and theoretical efforts.

## 2. Experimental Studies

In general, it is possible to produce molecular dications by the following main three ways, in which the specific reactions indicated are only some of the examples that can be reported:electron impact:        MgBr + e^−^ → MgBr^2+^ + 3e^−^,(13)ion molecule reactions:        Ar^2+^ + N_2_ → ArN^2+^ + N,(14)double photoionization:        CO_2_ + hν → CO_2_^2+^ + 2e^−^.(15)

Our research group in the last decades has been involved in all three of the above experimental methods whose detailed descriptions are reported in the following subsections.

### 2.1. The Alkaline Earth Monohalide Dications Production by Electron Impact: The Stanford Experiment

Our experimental work on molecular dications started in 1995 at the Zarelab in the Department of Chemistry of the University of Stanford, USA. At that time, the alkaline earth monohalide dications were predicted to be detectable species in gas phase by theoretical calculations performed by Wright and coworkers [26,59], who demonstrated that they are thermodynamically stable doubly charged molecular ions. During a “Zarelab group meeting” on December 1994, Dick Zare suggested to try to produce and detect such dications by electron impact mass spectrometry in order to demonstrate their existence and verify the theoretical predictions by Wright and coworkers [26,62].

The experiment was done in Spring 1995. Several alkaline earth monohalides were generated by an oven working at high-temperature (up to 1550 K) as an effusive beam containing alkaline earth atoms (M) and monohalide (MX) molecules (where M = Mg, Ca, Sr, Ba and X = F, Cl, Br, I) in a molecular beam apparatus working at a pressure of about 10^−7^–10^−8^ torr. This beam was obtained by mixing approximately equimolar amounts of the alkaline earth metal and the corresponding dihalide salt (MX_2_) and raising the oven temperature above their melting points. The production of the monohalide (MX) precursor molecules was obtained by the following exothermic gas-phase reaction:M + MX_2_ → 2MX.(16)

After that, the effusive beam containing a mixing of MX and M was directed into an Extrel quadrupole mass spectrometer (200 mm ¬ 9 ± 5 mm rods) where atomic and molecular species were ionized by an Extrel ionizer (model 020-2) via electron impact. In such a way, the diatomic dications were generated by electron bombardment:MX + e^−^ → MX^2+^ + 3e^−^,(17)
and detected with a channeltron electron multiplier (Galileo, model 4816) obtaining mass spectra at high resolution (the pressure inside the mass spectrometer chamber was maintained at about 10^−8^ torr) for the BaX^2+^ (X = F, Cl, Br, I), MgBr^2+^, CaBr^2+^, and SrCl^2+^ [63], demonstrating their existence. In Figure 3a,b are reported a scheme and a picture of the used experimental device, respectively, whereas in Figure 3c the obtained mass spectra can be appreciated.

### 2.2. The Production of Metastable Dications by Ion-molecules Reactions: The Trento Experiment

After this first experiment, we started a collaboration with the research group of D. Bassi and P. Tosi at the Physics Department of the University of Trento (Italy) in 1997. The scientific project was aimed at our proposal to produce for the first time a molecular dication via the following two strategies: (i) by Penning ionization of the monocations and (ii) by ion-molecule reaction. 

The first attempt failed but the second was successful, and we were able to study the first ion-molecule reaction involving an atomic dication and producing a doubly charged metastable molecule. The generated molecular dication was ArN^2+^, a metastable species able to release a considerable amount of KER (about 5 eV) by the reaction (14) for which the cross section as a function of the collision energy was measured (see Figure 4), indicating a maximum at a collision energy of about 10 eV [64]. The experiment was done by a molecular beam apparatus using an ion-molecule reaction mass spectrometer. A beam of Ar^2+^ ions was produced after electron bombardment of Ar with subsequent mass selection. Such an ionic beam was injected into a radio-frequency octopole ion guide mounted inside a scattering cell, where the N_2_ reactant gas was introduced at a pressure lower than 10^−4^ mbar in order to guarantee the possibility to study the reactive event in a single collision condition. The ion energy was varied by changing the octopole dc potential in order to evaluate the relative collision energy. Ionic reactant and final products were collected, focused and guided to a quadrupole mass filter for their selection and counting. A scheme of the experimental device as well as a picture of the molecular beam machine are reported in Figure 4, together with the measured reactive cross section as a function of the collision energy.

After this pioneering experiment, the laboratory of Trento was able to continue such kind of studies, and other analogous ion molecule reactions producing molecular dications containing noble gas atoms were investigated as, for example, the one forming ArC^2+^ [65].

### 2.3. Molecular Dications by Double Photoionization Using Synchrotron Radiation: The Elettra Experiments

From 2001, the Perugia research group started to explore the third method in generating molecular dications by double photoionization experiments (see for example reaction (15) above) using a tunable and very intense UV and EUV photon source as the synchrotron radiation ones available at the GasPhase and CiPo beamlines of Elettra Synchrotron Facility of Basovizza (Trieste, Italy). 

The first experiment was done in 2001 at the GasPhase Photoemission beamline of Elettra using the ARPES (Angular Resolved PhotoEmission Spectroscopy) end station. It concerned the double photoionization of HBr molecules, by synchrotron radiation in the energy range between 25 and 55 eV via a mass spectrometric determination. The HBr^2+^ and Br^2+^ product ions were detected by the photoelectron-photoion-coincidence technique, while the H^+^ + Br^+^ formation, which follows the double ionization of HBr, was studied by photoelectron-photoion-photoion-coincidence technique. The HBr^2+^ dications were produced with a threshold energy of 32.4 ± 0.4 eV (see Figure 5a), while the dissociative channel leading to H^+^ + Br^+^, showed a threshold at around 33.0 eV. The production of H + Br^2+^ occurred with a threshold energy of 40.2 ± 0.4 eV [66]. A scheme of the used experimental device is reported in Figure 5b.

Subsequently, analogous experiments were carried out starting from the neutral precursors HCl and HI, generating their respective molecular dications HCl^2+^ and HI^2+^ and measuring the energy thresholds for their formation as well as for the fragmentation channels following the Coulomb explosion fragmentation [67,68,69]. It has to be noted that in the case of HBr, we were able to perform a high-resolution experiment (≈ 10 meV) by detecting, in coincidence, the two electrons emitted with nearly zero kinetic energy using the TPEsCO (threshold photon-electron coincidence) technique. In such a way, in the Summer of 2003, very detailed information about the structure of the HBr^2+^ dication was obtained, as for example: (i) the vibrational energy spacing of the lowest-lying X ^3^Σ^−^_1,0_ and a ^1^Δ_2_ electronic states; (ii) the energetic location of the excited b ^1^Σ^+^ state; and (iii) high resolution TPEsCO spectra in which the two spin–orbit components in the case of the ground electronic ^3^Σ state were well resolved [70]. These experiments allowed our group to fully characterize the interaction components describing the potential energy curves for the hydrogen halide dications family HX^2+^ (where X = F, Cl, Br, I), by combining results from a semiempirical method with ab initio calculations [71].

After that, since 2007 our interest was addressed to simple molecular dications, which are interesting from an atmospheric point of view and in astrochemistry. Since many of such ionic species were expected to be metastable, we decided to concentrate our efforts on the measure of their lifetime and energy threshold formation as well as the KER and angular distribution determinations of ionic fragments coming out from their Coulomb explosion. This goal was achieved performing double photoionization experiments with tunable synchrotron light and using an ion position sensitive detector developed by Lavollée [72] and allowing electron-ion-ion coincidences measurements. By such a detector, shown in Figure 6a,b, we were able to measure the spatial momentum components of the dissociation ionic products, since it couples a time-of-flight (TOF) spectrometer equipped and an ion imaging detector (stack of three micro-channel-plates MCP with a multi-anode array arranged in 32 rows and 32 columns) [72]. This device, still fully working today, allows the measurement of coincidence spectra of produced ions as a function of the investigated photon energy (see for example Figure 6c) and, consequently, the relative cross section for all investigated dissociation channels and the KER of the ion products employing the procedure suggested by Lundqvist et al. [73] can be extracted. Furthermore, through the analysis of coincidences distribution as a function of the arrival time differences (t_2_ − t_1_) of fragment ions to the ion imaging detector, it is possible to determine the lifetime of the metastable intermediate molecular dications adopting the method proposed by Field and Eland [74]. Finally, the angular distributions of the fragment ion products with the extraction of the related anisotropy parameter according to the procedure first introduced by Zare [75,76] can be obtained. 

By this technique, we were able to produce and characterize several molecular dications such a CO_2_^2+^ [77,78], N_2_O^2+^ [79,80], C_6_H_6_^2+^ [81,82], and C_2_H_2_^2+^ [83,84], and more recently C_3_H_6_O^2+^ [85,86], C_2_H_5_NO^2+^ [87] (the dications of methyloxirane and N-methylformamide, respectively), pointing out their important role in the upper atmosphere of planets of the Solar system. In all cases, we performed double photoionization experiments using the tunable linearly polarized synchrotron radiation (with photon energy resolution of about 1.5 meV) as it is available at the GasPhase and CiPo beamlines of Elettra measuring: (i) the threshold energy for the different open fragmentation channels; (ii) the relative cross sections; (iii) the KER distribution of fragment ions at different photon energies; (iv) the angular distributions of fragment final ions with the relative anisotropy parameter β [75,76]. As an example of this kind of experimental observable obtainable in our experiments, Figure 6d and Figure 7 report on few data collected in the case of the double photoionization experiments performed with CO_2_ [77,78], while Figure 8 shows the analogous data obtained with N_2_O [79,80].

In particular, Figure 8a reports on the high anisotropy, which characterizes the microscopic fragmentation dynamics following the Coulomb explosion of the metastable N_2_O^2+^ dication formed by the double photoionization in the 30–50 eV photon energy range. In this case, we were able to perform a detailed analysis of the measured angular distribution of the ionic products generated by the two recorded two-body fragmentation channels below: N_2_O + hν → N_2_O^2+^ + 2e^−^ → N^+^ + NO^+^ + 2e^−^(18)N_2_O + hν → N_2_O^2+^ + 2e^−^ → N_2_^+^ + O^+^ + 2e^−^(19)

In a wide range of the investigated photon energy (35–50 eV), the recorded angular distributions for both ion pair products of reactions (18) and (19) are characterized by a strong anisotropy (see for example the data in Figure 8a relative to a photon energy of 36 eV). Indeed, the fitting procedure of the recorded angular distributions according to the method by Zare [75,76] gave an anisotropy parameter of β = 1.5 for fragmentation channels above, indicating that the two pairs N^+^/NO^+^ and N_2_^+^/O^+^ of product ions are emitted by the Coulomb explosion of the intermediate N_2_O^2+^ along a direction parallel to the polarization vector of the synchrotron radiation (i.e., on the plane perpendicular to the light beam direction). As discussed in detail in [80], this is a clear indication that the VUV photon absorption by N_2_O molecule is strongly anisotropic and the lifetime of the metastable N_2_O^2+^ dication against the dissociation into two singly charged ions is comparable or shorter than 10^−11^ s. Furthermore, our data point out that, above the vertical threshold for the N_2_O^2+^ formation (35.8 eV), the two fragment ions dissociate with a KER that remains unchanged (being about 6.2 eV for N^+^ + NO^+^ and 5.0 eV for O^+^ + N_2_^+^) as the photon energy increases [80]. Figure 8b shows the recorded KER distributions for the ion pair products of both fragmentation reactions (18) and (19) at different investigated photon energies.

#### Looking at the Escape of Ions from Planetary Atmospheres

A very important characteristic for the chemistry of molecules in the interstellar medium and planetary ionospheres is that they interact with the electromagnetic waves: γ and X rays, UV light. For example, the interaction with the ultraviolet light is responsible of the limited growth of the organic molecules. Our experiments, discussed in the previous section, demonstrated that using the UV light, molecules can be ionized producing mono and dications. Moreover, as mentioned in the Introduction section, considerable amounts of simple doubly charged species (as for example CO_2_^2+^ and N_2_^2+^) have been detected in Mars, Venus and Titan ionospheres [4,5,6]. Furthermore, also N_2_O^2+^ has been suggested to be present in the upper atmosphere of Titan in a detectable amount by Dobrijevic et al. [88], who considered for the first time the coupling between nitrogen and oxygen chemistry in that environment.

In the case of CO_2_, our experiments performed at the Elettra Synchrotron Facility using a 34–50 eV photon energy, whose results are summarized in the previous section (see Figure 6 and Figure 7), the following molecular fragmentation has been observed:CO_2_ + hν → CO_2_^2+^ + 2 e^−^ → CO^+^ + O^+^ + 2e^−^(20)with a total kinetic energy of product ions ranging between 2 and 6 eV (see Figure 6d). Since the CO_2_ is involved in several atmospheric phenomena of the Earth and of other planets like Mars and Venus, where it is the main important chemical component, and because the presence in these environments of ionizing radiations is able to induce double photoionization processes, the reaction (20) was widely studied by our research group [77,78]. Following the interesting discussion that opened in the early 2000s in the scientific community concerning the possible role of molecular dications in the upper atmosphere of planets [7,8,9], we were able to propose an explanation of the lack in the O^+^ expected concentration of the Mars atmosphere, as measured by MARINER 6 spacecraft and VIKING 1 lander, invoking the atmospheric escape of O^+^ by dissociative double photoionization of CO_2_ molecules via Coulomb explosion of the metastable intermediate CO_2_^2+^ dication. [89,90]. This is a general consequence of the exotic behavior of the metastable doubly charged molecular ions (as for example CO_2_^2+^, N_2_O^2+^, N_2_^2+^, C_2_H_2_^2+^, etc.) when they are formed in planetary atmospheres with the possibility of creation of dissociative ionic products with a very high kinetic energy content of several eV. In the case of CO_2_, the production of CO^+^ and O^+^ fragments with a KER of about 2.0 and 3.8 eV, respectively, as measured in our experiments [78] (see Figure 6d) can explain the lack in the O^+^ expected concentration of the Mars atmosphere as discussed in detail in [90]. This translational energy, in fact, is large enough, in the case of Mars and Titan, to allow these ionic fragments to reach sufficient velocity in order to escape into space as it can be seen from data reported in Table 1. Therefore, this process can in principle contribute to the continuous erosion of the atmospheres of such planets 

This important observation is not limited to carbon dioxide but can be extended to other important molecules of interest in atmospheric and astrochemistry phenomena. This is the case of the nitrous oxide, acetylene, benzene, propylene oxide (the first chiral molecule recently discovered in space [91]) and N-methylformamide double UV photoionization in in the photon energy range of 28–65 eV: The generation of the relative intermediate dications and the subsequent fragmentation into O^+^, CO^+^, N^+^, CH_3_^+^, CH_2_^+^, CH^+^, C^+^, and H^+^ which are characterized by a translational energy ranging between 1.0 and 6.0 eV, indicates that such an energy content is large enough to allow their escape process from the upper atmospheres of Mars and Titan, as it is shown in Table 1. Further studies are planned and in progress in our laboratory to clarify important details about the chemistry of planetary atmospheres involving excited metastable species [92], chiral molecules, neutral reactive species [93], and radicals as ClNO.

## 3. Theoretical and Computational Calculations

As a prototype of dication, let us consider the benzene dication [C_6_H_6_]^2+^, which was investigated several times through the years by our group in different contexts [81,82,94]. Benzene is an important molecule in astrochemical environments since it is a precursor of polycyclic aromatic hydrocarbons (PAH), important for the understanding of many processes happening in the interstellar medium (ISM) as, for instance, the identification of IR bands observed in a number of astrophysical sources [95,96,97] or in the heating and cooling of the ISM [98,99]. Benzene is an important molecule also in the atmosphere of Titan, the massive moon of Saturn. This atmosphere is reminiscent of the primeval atmosphere of Earth and, therefore, its investigation should provide important information for understanding the origin of life on Earth [100,101]. Among the species identified by Cassini Ion Neutral Spectrometer (INMS) in the upper atmosphere of Titan [102], benzene shows a relatively important mole fraction, being 1.3 × 10^−6^ at 950 km [103], while solid benzene has been identified on the surface of Titan [104]. Benzene dication should dissociate through several reactive channels providing charged fragments. These reactive channels should be exothermic because of the Coulomb repulsion between charged fragments. However, these reactions shows barriers, which suggest that Coulomb explosion is not a significant channel under most astrophysical conditions, i.e., low temperatures and low pressures.

The potential energy surfaces of interest were investigated by locating the lowest stationary points at the B3LYP level of theory [105,106], in conjunction with the 6-311+G(*d*) basis set [107,108]. At the same level of theory, we have computed the harmonic vibrational frequencies in order to check the nature of the stationary points, i.e., minimum if all the frequencies are real, saddle point if there is one, and only one, imaginary frequency. The assignment of the saddle points was performed using intrinsic reaction coordinate (IRC) calculations [109,110]. The geometry of all the species was optimized without any symmetry constraints considering for all the investigated species the electronic ground state. For all the stationary points, the energy was computed also at the higher level of calculation CCSD(T) [111,112,113] using the 6-311++G(3*df*,3*pd*) basis set [107,108], following a well-established computational scheme [114,115,116,117,118,119,120,121]. Both the B3LYP and the CCSD(T) energies were corrected to 0 K by adding the zero point energy correction computed using the scaled harmonic vibrational frequencies evaluated at B3LYP level. 

The main dissociative processes of C_6_H_6_^2+^, leading to the ion pairs reported in the following reactions, have been previously characterized by our group by exploiting a photoelectron–photoion–photoion coincidence technique [81,82].C_6_H_6_^2+^ → C_3_H_3_^+^ + C_3_H_3_^+^      −3.23 eV(21)C_6_H_6_^2+^ → CH_3_^+^ + C_5_H_3_^+^      −0.77 eV(22)C_6_H_6_^2+^ → C_2_H_3_^+^ + C_4_H_3_^+^      −0.62 eV(23)C_6_H_6_^2+^→ C_2_H_2_^+^ + C_4_H_4_^+^      +0.31 eV(24)

The Δ*H*_0_ values computed at the CCSD(T)/6-311++G(3df,2pd)//B3LYP/6-311+G(d) level at 0 K are as well reported. We notice that while the dissociations into C_3_H_3_^+^ + C_3_H_3_^+^, CH_3_^+^ + C_5_H_3_^+^, and C_2_H_3_^+^ + C_4_H_3_^+^ are exothermic reactions at 0 K, the dissociation of C_6_H_6_^2+^ into C_2_H_2_^+^ + C_4_H_4_^+^ ion pairs is an endothermic process at 0 K. Reaction (24), therefore, will never proceed under the very unfavorable conditions of ISM or the atmosphere of planets like Titan, i.e., very low temperature and pressure. We will not consider, therefore, reaction (24) anymore. Reactions (22), (23), and mostly (21) could proceed also in the ISM medium if these reactions do not imply any energy barrier. In order to clarify this aspect, we have computed the potential energy surface (PES) of the dissociation of C_6_H_6_^2+^ into the ion pairs present in reactions (21), (22), and (23). Figure 9 reports the PES for these reactions computed at CCSD(T)/6-311++G(3*df*,3*pd*)//B3LYP/6-311+G(*d*) level. In red we have reported the scheme for reaction (21), in blue for reaction (22) and in green for reaction (23), while the common part is in black. We can notice that the first step for the three reactions is the opening of the C_6_H_6_^2+^ ring, which implies the overcoming of a barrier of 2.70 eV. This is a barrier too high to be overcome in the ISM medium or in the extreme conditions of astrophysical objects. Therefore, this is the reason why a dication such as C_6_H_6_^2+^ can have a very long lifetime.

Let us consider the pathways leading to the ion pairs products for the three reaction (21), (22), and (23). For reaction (21), the first step is the opening of the benzene ring, which implies a barrier as high as 2.70 eV. After this barrier, we have the formation of a minimum about 0.65 eV more stable than the saddle point, which dissociates directly into the two C_3_H_3_^+^ ions. However, this dissociation requires the overcoming of a barrier energy of 1.45 eV. Although the whole reaction is exothermic by 3.23 eV, the presence of the very high transition states makes it unfeasible. The first step of the pathway leading to the products of reaction (22) is the same of reaction (21), i.e., the opening of the ring, which implies a barrier of 2.70 eV. Then the minimum [CHCHCHCHCHCH]^+2^ can isomerize and, passing through six minima and seven transition states, can give the ion pairs CH_3_^+^ + C_5_H_3_^+^ products. In Figure 9, we can notice that two saddle points (at 1.92 and 1.22 eV) show an energy slightly lower than the reactants. This is due to the fact that we optimized the stationary points at B3LYP/6-311+G(*d*) level and we refined the energetics computing the energy of the stationary points at fixed geometry at CCSD(T)/6-311+G(3*df*,3*pd*) level. These saddle points are true transition states at the level of the geometry optimization. Also, for reaction (22), the products lie at an energy lower than the reactants, but the very high initial energy barrier makes also this reaction unfeasible under extreme astrophysical conditions. The pathway leading to the products of reaction (23) shows the first part in common with the pathway of reaction (22), until the formation of the minimum [CH_2_CHCHCCCH_2_]^+2^. This species can isomerize to [CH_3_CHCCCCH_2_]^+2^ or dissociate straight to the ion pairs C_4_H_3_^+^ + C_2_H_3_^+^, through an energy barrier as high as 2.19 eV. Also channel (23), therefore, is unfeasible. 

In Figure 10, we have reported the optimized geometry of C_6_H_6_^2+^. Its structure is not planar, because of the double charge, but it is slightly distorted towards a chair conformation. The breaking of a C–C bond leads to the transition state, whose geometry is as well reported in Figure 10, which is the first step for all the considered reactions. The breaking of this C–C bond requires enough energy to avoid the Coulomb explosion of C_6_H_6_^2+^.

We have seen that the C_6_H_6_^2+^ dication is stabilized under astrophysical conditions since its dissociation into ion pairs implies strong geometrical rearrangments and, therefore, high energy barriers. However, also very simple diatomic dications can avoid the Coulomb explosion since their ground state is a metastable state. We studied this aspect many years ago for the diatomic Be_2_^2+^ system, whose ground state potential supports 13 vibrational levels with long lifetimes with respect to unimolecular decay [122,123].

## Figures and Tables

**Figure 1 molecules-25-04157-f001:**
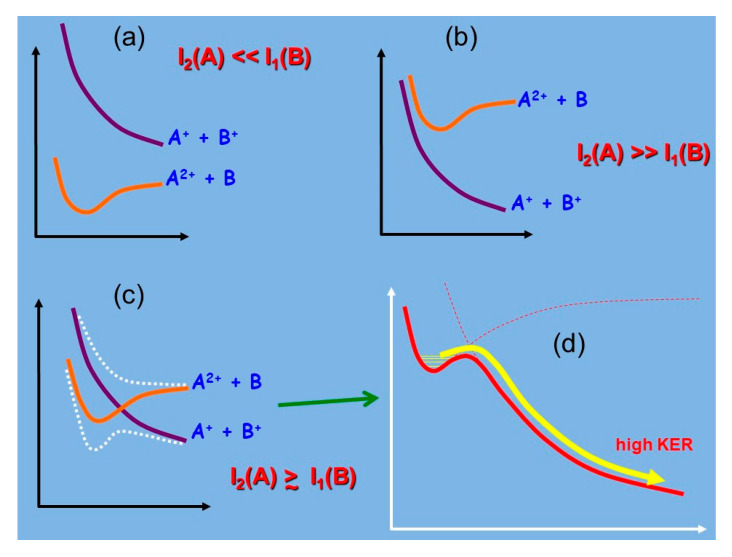
Scheme of the potential energy curves for AB^2+^ diatomic molecular dications: (**a**) the case of the thermodynamically stable species; (**b**) unstable species; (**c**) the metastable dications resulting from the avoided crossing of the involved potential energy curves; (**d**) the potential energy curve coming out from case (**c**) producing A^+^ and B^+^ fragments with high kinetic energy release (KER) by Coulomb explosion. In all panels, the interatomic distance is shown in the x axis while the energy is shown in the y axis.

**Figure 2 molecules-25-04157-f002:**
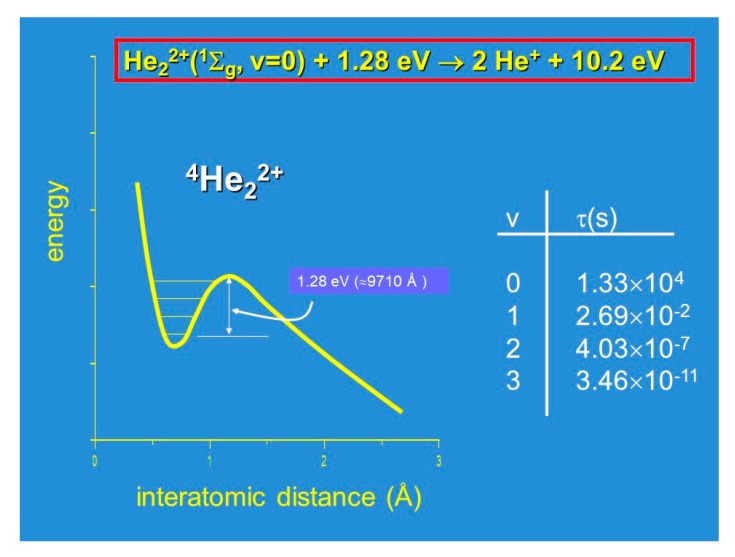
Scheme of the potential energy curve describing the interaction in the simplest He^2+^ metastable molecular dication studied first by Pauling in 1933 [18].

**Figure 3 molecules-25-04157-f003:**
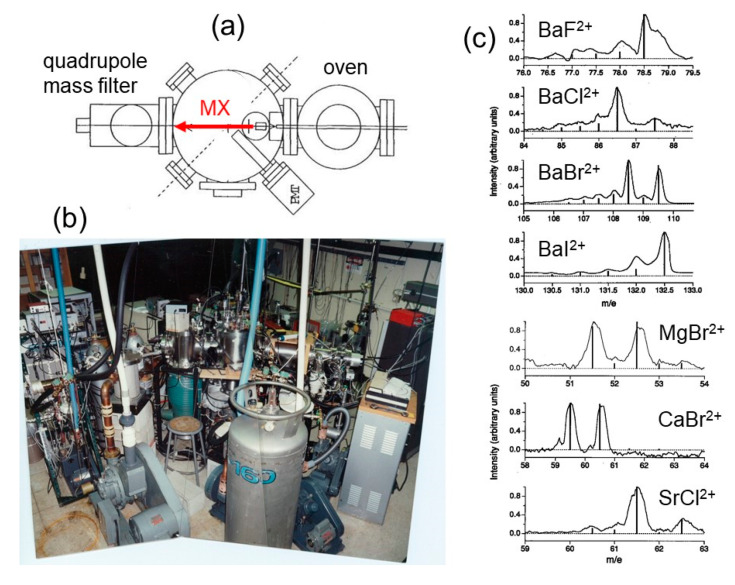
The molecular beam machine used in the Stanford experiment for the first generation of the alkaline earth monohalide dications in 1995: (**a**) a scheme of the experimental set up; (**b**) a picture of the apparatus that was working in the Zarelab; (**c**) the mass spectra obtained in the generation of BaX^2+^ (X = F, Cl, Br, I), MgBr^2+^, CaBr^2+^, and SrCl^2+^ molecular dications [63].

**Figure 4 molecules-25-04157-f004:**
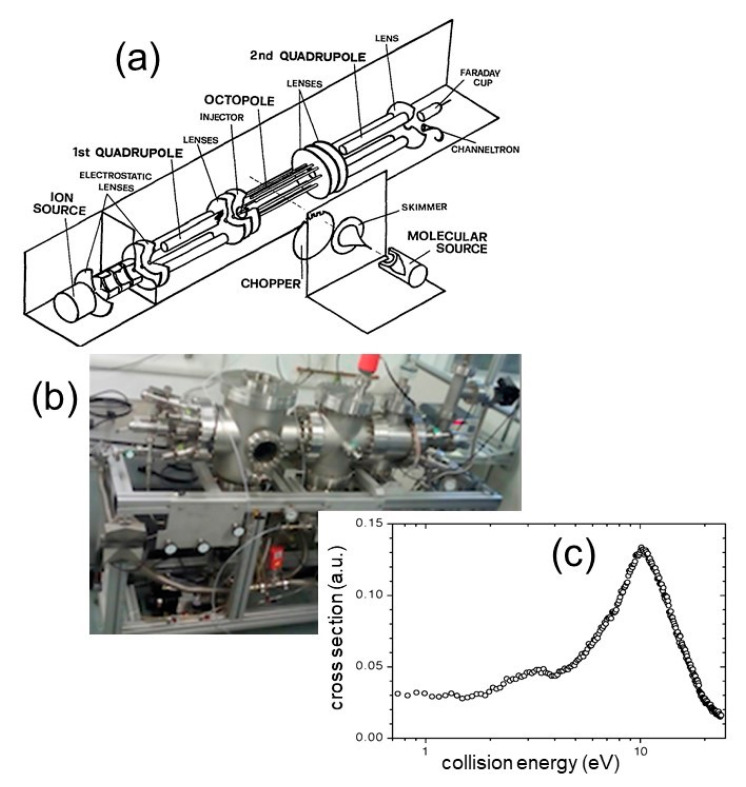
The ion guide molecular beam apparatus used in the Trento experiment on 1998 for the first production of molecular dications containing rare gases by ion molecule reaction: (**a**) a scheme of the experiment; (**b**) a picture of the machine; (**c**) the cross section for the studied reaction Ar^2+^ + N_2_ → ArN^2+^ + N as a function of the collision energy [64].

**Figure 5 molecules-25-04157-f005:**
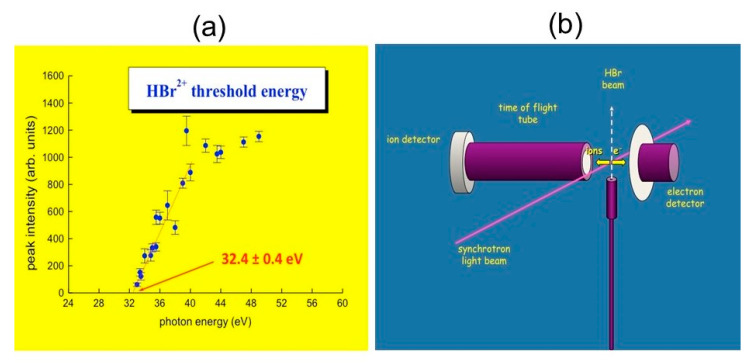
The double photoionization of HBr using synchrotron radiation in the 25–55 eV photon energy range: (**a**) mass spectrometric determination of the threshold energy for the production of HBr^2+^ molecular dication; (**b**) a scheme of the set up used in our first experiment at Elettra Synchrotron Facility of Trieste on 2001 [66].

**Figure 6 molecules-25-04157-f006:**
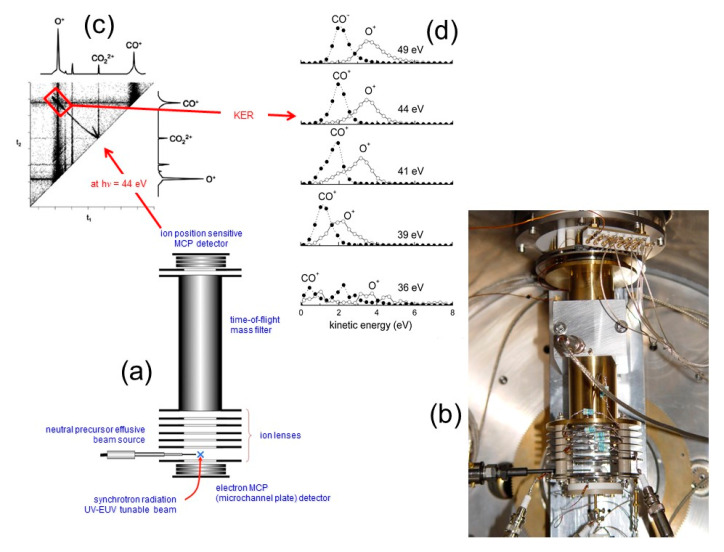
The PEPIPICO (photoelectron–photoion–photoion coincidence) detector coupled with TOF mass spectrometer especially designed to measure the spatial momentum components of product fragments ions in the double photoionization experiments [72]: a scheme (**a**) and a picture (**b**) of the device; (**c**) an example of a typical recorded coincidence spectrum with the relative mass spectra collected in the double photoionization experiment of carbon dioxide at a photon energy of 44 eV. In red color is pointed out the region of the coincidences signal used for the evaluation of the KER distributions for CO^+^ and O^+^ fragment ions generated from the Coulomb explosion of the CO^2+^ metastable dication; (**d**) the KER distributions for CO^+^/O^+^ pair of fragment ions as a function of the investigated photon energy [77] indicating that O^+^ ions are produced with a high kinetic energy content ranging between 2.2–3.8 eV.

**Figure 7 molecules-25-04157-f007:**
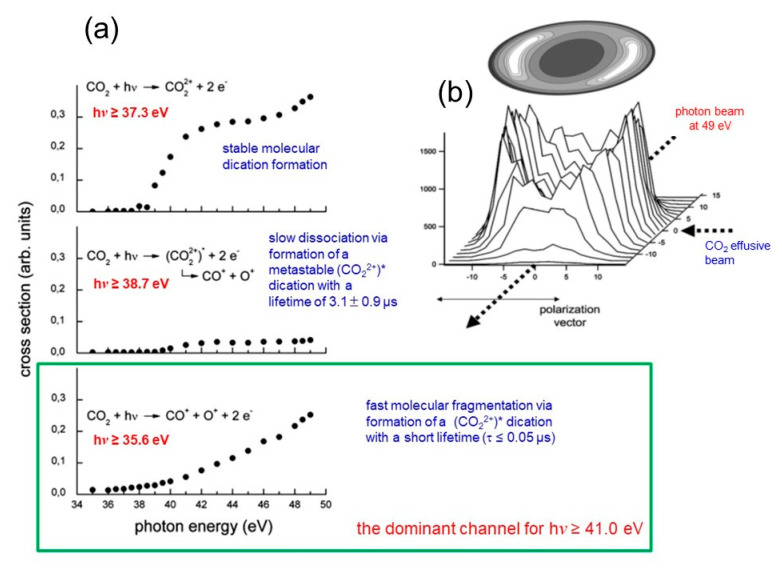
Some typical data obtainable in the double photoionization experiments performed using the PEPIPICO device of Figure 6: (**a**) the relative cross sections as a function of the investigated photon energy obtained in the case of CO_2_ [77]; in this experiment, two different metastable CO_2_^2+^ dications were detected and the relative lifetimes were measured (for details see ref. [77]); (**b**) an example of the typical 3D representation of the angular distribution for CO^+^/O^+^ ions products recorded at a photon energy of 49 eV [78].

**Figure 8 molecules-25-04157-f008:**
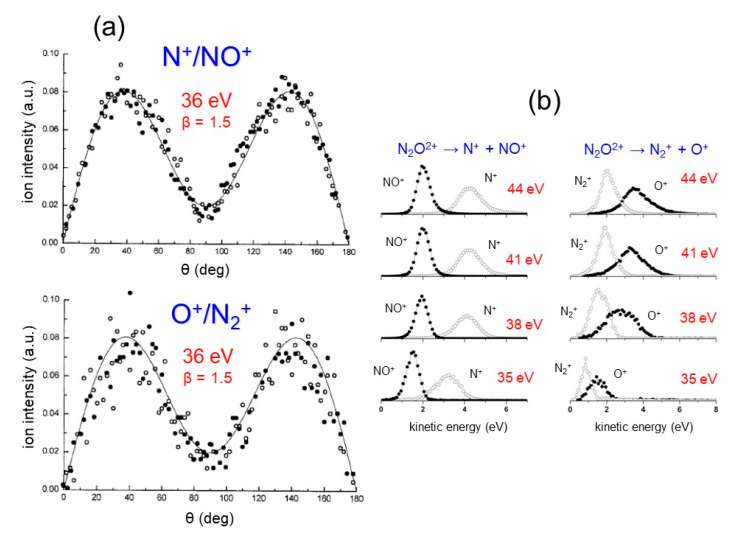
Data from the double photoionization of N_2_O by synchrotron radiation in the 30–50 photon energy range: (**a**) the angular distributions obtained for the two fragmentation channels (see Equations (17) and (18) in the text) at a photon energy of 36 eV [80]; the black points refer to the diatomic ions NO^+^ or N_2_^+^ and the open circles to the atomic N^+^ or O^+^ ones; the best fitting of the experimental data allowed the determination of the anisotropy parameter β (see text); (**b**) the KER distributions for single product ions of reactions (17) and (18) occurring by the Coulomb explosion of the metastable N_2_O^2+^ molecular dication [80].

**Figure 9 molecules-25-04157-f009:**
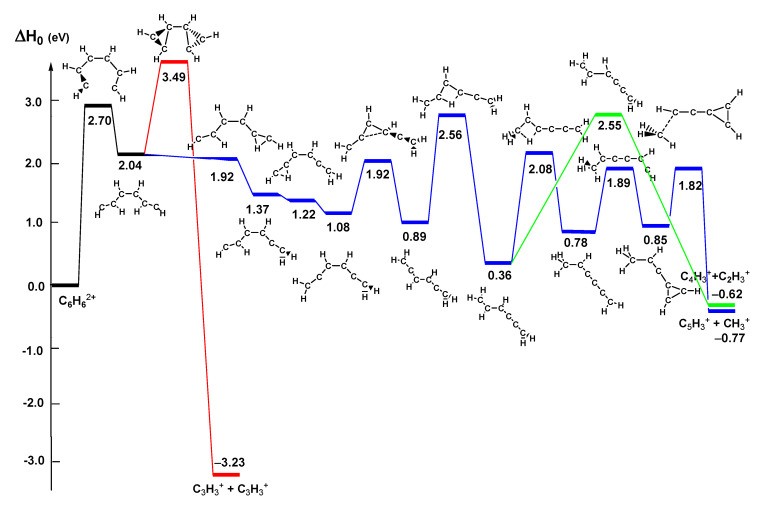
Potential energy surface of the selected dissociation channels of C_6_H_6_^2+^ computed at CCSD(T)/6-311++G(3*df*,3*pd*)//B3LYP/6-311+G(*d*) level. Relative energies (eV) at 0 K with inclusion of zero point energy correction with respect to C_6_H_6_^2+^.

**Figure 10 molecules-25-04157-f010:**
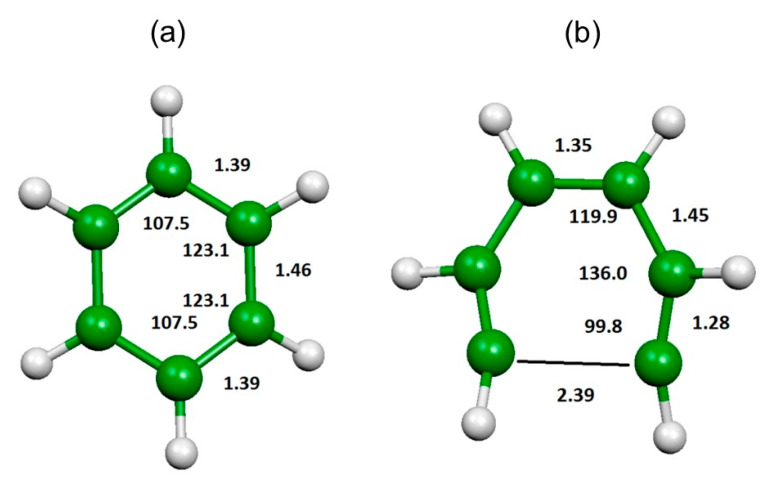
Optimized geometries computed at the B3LYP/6-311+G(*d*) level of C_6_H_6_^+2^ (**a**) and the first transition state (**b**) reported in Figure 9. Bond lengths in Å, angles in degrees.

**Table 1 molecules-25-04157-t001:** The measured KER (eV) for some ionic species generated by Coulomb explosion of C_2_H_2_^2+^, CO_2_^2+^, N_2_O^2+^ and C_2_H_5_NO^2+^ (from N-methylformamide) metastable dications of astrochemical interest compared with their escape energy (eV) from the atmosphere of various planets of the Solar system and of Titan (the largest moon of Saturn) evaluated at the exobase [7].

Ions	Measured KER Range (eV)	Escape Energy (eV) from the Atmosphere of Some Planets			
		Earth	Venus	Mars	Titan
H^+^	2.8–6.0 ^1^; 2.8–6.0 ^2^	0.62	0.53	0.13	0.02
C^+^	1.7–3.3 ^1.^	7.4	6.4	1.5	0.28
CH^+^	1.3–3.2 ^1.^	8.0	6.9	1.6	0.30
CH_2_^+^	1.6–2.9 ^1.^	8.6	7.5	1.8	0.32
N^+^	2.3–5.2 ^3.^	8.6	7.5	1.8	0.32
CH_3_^+^	2.7–4.8 ^2.^	9.2	8.1	1.9	0.34
O^+^	1.0–5.2 ^3^; 2.2–3.8 ^4.^	9.8	8.6	2.0	0.37
C_2_H^+^	0.1–0.4 ^1.^	15.4	13.3	3.1	0.58
CO^+^	0.4–2.6 ^4.^	17.3	14.9	3.5	0.65
N_2_^+^	0.4–2.9 ^3.^	17.3	14.9	3.5	0.65
NO^+^	1.0–2.9 ^3.^	18.5	16.0	3.75	0.70

^1^ from references [78]; ^2.^ from references [87]; ^3^ from references [80]; ^4.^ from references [84].

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
