# Peer review of "Production and Characterization of Molecular Dications: Experimental and Theoretical Efforts"

_molecules, 2020, doi:10.3390/molecules25184157_

Round 1

Reviewer 1 Report

This is an interesting review in energetic materials. mechanism and historical evolution of the field is presented. it is a good read and it is connected to astro-chemistry/physics. It would be very useful to include the latest developments in high intensity laser experiments where multiply ionised atoms and molecules can be created by strong field action. 

It would be also important to add discussion on Coulomb explosion as function of material/gas density. It is important to define density criteria when Coulomb explosion becomes impposible due to limitation of electron removal. 

Author Response

Point-by-point responses to the reviewers’ comments - manuscript ID molecules_929299

We thank you and both reviewers #1, and #2 for their positive comments and useful suggestions aimed to improve the quality of the manuscript ID molecules_929299.

We made a revision of the manuscript, addressing all the comments and suggestions by the reviewers.

All made changes in the manuscript are highlighted in red color in its revised version (see new uploaded revised manuscript) and are listed below.

Responses to the Reviewer #1 comments:

Reviewer #1 – general comment: This is an interesting review in energetic materials. mechanism and historical evolution of the field is presented. it is a good read and it is connected to astro-chemistry/physics.”.

Author reply: We thank sincerely the reviewer #1 for his very positive comments.

Reviewer #1 - addressed point 1: It would be very useful to include the latest developments in high intensity laser experiments where multiply ionised atoms and molecules can be created by strong field action.”.

Author reply and made modifications: We agree with the reviewer #1 and we thank him for his suggestion. In order to address this point, we added the following sentence at page 4, lines 127-129 of the “Introduction” section with related new 3 references: “Remarkable recent developments have been done in the production and characterization of multiply ionized molecular species using high intensity laser experiments, as intense femtosecond lasers and X-ray free electron lasers (see for example refs. 59-61 and references therein).”;

The new added references are the followings:

  1. Hankin, S. M.; Villeneuve, D. M.; Corkum, P. B.; Rayner, D. M. Intense-field laser ionization rates in atoms and molecules. Phys. Rev. A 2001, 64, 013405.
  2. Agostini, P. and DiMauro, L. F. Chapter 3 - Atomic and Molecular Ionization Dynamics in Strong Laser Fields: From Optical to X-rays. Adv. At. Mol. Opt. Phy. 2012, 61, 117-158.
  3. Yatsuhashi, T. and Nakashima, N. Multiple ionization and Coulomb explosion of molecules, molecular complexes, clusters and solid surfaces. J. Photochem. Photobiol. C: Photochem. Rev. 2018, 34, 52–84.

Reviewer #1 – addressed point 2: It would be also important to add discussion on Coulomb explosion as function of material/gas density. It is important to define density criteria when Coulomb explosion becomes impossible due to limitation of electron removal..

Author reply and made modifications: we thank the reviewer for his comment and we added the following sentences at page 4, lines 129-136 of the revised version of the manuscript: “In particular, Yatsuhashi and Nakashima [61] in their recent review paper pointed out on important advances performed on Coulomb explosion imaging experiments. This kind of technique allowed the determination of the static structures for complex and chiral molecules, geometric isomers and molecular aggregates. Furthermore, the importance of Coulomb explosion of solids (i.e. metals, dielectrics, semiconductors, polymers and molecular aggregates) has grown in recent years and pioneering time-resolved studies of surface electric fields have been done clarifying the contribution of Coulomb explosion to the mechanism for ablation of solid surfaces [61].”.

Reviewer 2 Report

Such doubly charged molecules have been discussed for decades. In fact, I recall listening to a lecture on the topic more than thirty years ago, presented by a Greek scientist who was funded by the US Air Force in a program searching for high energy density materials (to be used as propellants or explosives). I note that Nicolaides' "volcanic states" are cited [19].
That presentation concerned mostly theoretical studies; I appreciate that the TR addresses both theoretical and experimental work.

The authors are based in Perugia (Italy) and reminisce about their earlier work at Stanford (CA, USA). While much of the text is quite readable, in various locations some Italian emphasis dominates over English grammar. It would be good to have somebody with sufficient physics understanding, but also a good feeling for the English language, edit the text.

Line 17, "authors' research groups" might be correct
Line 25, "a lot amount of energy" is incorrect English, "a high amount of energy" or "a lot of energy" would be better. "in flames", not "in the flames"
Line 28, "in 1996"
Line 34, insert comma after "Herman [10]"
Line 61 and figure caption 1: "KER" is the "kinetic energy release" (without a "d" at the end).
Line 65, "his" pioneering work
Line 143, "in" Spring
Line 192, the Perugia research group
Line 216, wrong genitive, delete " 's"
Line 334, Table 1 header, Titan is not a planet

These notes are just examples, not a complete list.

Most of the text is organized along the personal experience of the authors in their experiments. This may be less systematically structured than I would have expected, but the storyline and the many references have their own value.

After some moderate editing for language flow and grammar, this manuscript should be appropriate and acceptable for "Molecules".

Author Response

Point-by-point responses to the reviewers’ comments - manuscript ID molecules_929299

We thank both reviewers #1, and #2 for their positive comments and useful suggestions aimed to improve the quality of the manuscript ID molecules 929299.

We made a revision of the manuscript, addressing all the comments and suggestions by the reviewers.

All made changes in the manuscript are highlighted in red color in its revised version (see new uploaded revised manuscript) and are listed below.

Responses to the Reviewer #2 comments:

Reviewer #2 – general comment: Such doubly charged molecules have been discussed for decades. In fact, I recall listening to a lecture on the topic more than thirty years ago, presented by a Greek scientist who was funded by the US Air Force in a program searching for high energy density materials (to be used as propellants or explosives). I note that Nicolaides' "volcanic states" are cited [19]. That presentation concerned mostly theoretical studies; I appreciate that the TR addresses both theoretical and experimental work..

Author reply and made modifications: We are grateful to the reviewer #2 for his positive comments.  

Reviewer #2 - addressed point 1: “The authors are based in Perugia (Italy) and reminisce about their earlier work at Stanford (CA, USA). While much of the text is quite readable, in various locations some Italian emphasis dominates over English grammar. It would be good to have somebody with sufficient physics understanding, but also a good feeling for the English language, edit the text.”.

Author reply and made modifications: We thank the reviewer #2 for his suggestion. In order to improve the English style and quality of the manuscript we reread and revised the manuscript with the help of our native English-speaking colleague, correcting inaccuracies and errors. all changes requested and listed by the reviewer (at lines 17, 25, 28, 34, 61, 65, 143, 192, 216, 334 were accepted (see the uploaded revised version of the manuscript).

Reviewer #2 - addressed point 2: Most of the text is organized along the personal experience of the authors in their experiments. This may be less systematically structured than I would have expected, but the storyline and the many references have their own value. After some moderate editing for language flow and grammar, this manuscript should be appropriate and acceptable for "Molecules..

Author reply and made modifications: We understand the point of view of the reviewer #2 but we believed it was useful to highlight the gradual path made by our research group in conceiving different experimental and theoretical efforts capable of providing during the years more useful information on molecular dications by showing the logic that guided us.
